# Visitor’s Willingness to Pay for National Park Entrance Fees in China: Evidence from a Contingent Valuation Method

**DOI:** 10.3390/ijerph182413410

**Published:** 2021-12-20

**Authors:** Lin Song, Yi Xue, Yaqiong Jing, Jincan Zhang

**Affiliations:** 1School of Economics and Finance, Xi’an Jiaotong University, Xi’an 710061, China; songlin9023@xjtu.edu.cn (L.S.); xueyi004@stu.xjtu.edu.cn (Y.X.); 2Qinghai Vocational and Technical Institute of Animal Husbandry and Vet, Xining 812100, China; jyqhehexixi@126.com; 3School of Management Engineering, Zhengzhou University, Zhengzhou 450001, China

**Keywords:** national park, willingness to pay, contingent valuation, Qinling Mountains, entrance fee

## Abstract

China has been making efforts in nature conservation by developing a new national park system. Setting a fee-based entrance policy for the newly established national parks can be challenging without information on visitors’ willingness to pay (WTP). Thus, this study aims to evaluate visitors’ WTP entrance fees with a dataset of 1215 visitors collected in China’s planned Qinling National Park (QNP). Using the double bounded dichotomous choice format of the contingent valuation method, we obtained the mean WTP for the entrance fee of QNP of 200 yuan. Visitors’ demand becomes relatively inelastic at the recommended entrance fee of 160 yuan when the expected ticket revenue will reach its maximum of 13.8 billion yuan. Our study also indicates that visitors’ WTP is significantly influenced by their gender difference, education level, income, number of trips to natural attractions, duration of stay, the intention of a future visit, and concern about commercial exploitation. Our empirical study provides insights into developing effective pricing policies and appropriate marketing strategies for China’s new national park system.

## 1. Introduction

China, one of the most diverse and heritage-rich countries in the world, has established its protected area system with more than ten types, covering 18% of its land area and 4.1% of its sea area [1]. The problem is that different kinds of protected areas overlap with each other and have been governed by multiple departments simultaneously [2,3], which weakens their conservation efficiency. The idea of building a national park system was first put forward at the Third Plenary Session of the 18th Communist Party of China (CPC) Central Committee in 2013. Since January 2015, when the National Development and Reform Commission and other 13 ministries jointly issued *the Scheme for Establishing National Park System Pilot*, China has launched ten pilot national parks involving 12 provinces. These pilot areas cover a total of nearly 0.22 million square kilometres, accounting for 3% of China’s land area [2]. Unlike previously established nature reserves that protect specific species, China’s national parks emphasise the authentic and integral protection of the natural ecosystems and natural and cultural heritage, as well as facilitate cultural, scientific, educational, recreational, and visiting opportunities.

The establishment and subsequent management of national parks needs a large amount of funding. Entrance fees have been an essential source of revenue used to improve the visitors’ experience in the protected areas of China [4]. Although the debate in charging fees for national parks has been a hot topic in both the media and the literature [5,6], the advantages of using entrance fees for national parks has been widely recognised in the literature [7], such as for generating revenues, reducing congestion, and promoting comparative equity [8]. Before embarking upon any fee-paying policy, national park authorities are suggested to solicit visitors’ willingness to pay (WTP) for the park to gain public support for the fee-paying [9,10,11] and set the optimal price for high revenues. The contingent valuation method (CVM) has been a well-recognised valuation method to estimate visitors’ WTP for the national parks in different countries [5,6,7,8,9,10,11,12,13]. A recent study by Halkos et al. [14] reviewed a large amount of CVM studies for parks, whereas the corresponding empirical studies ins China’s national parks have so far been limited. 

Due to the long-term sustainable financing need of China’s newly established national parks, this study aims to evaluate visitors’ WTP for an admission ticket to access the national park, in the hope that it may provide insights for policymakers to set sounder pricing policies. A double bounded dichotomous choice (DBDC) format of the CVM, recommended by the National Oceanic and Atmospheric Administration panel [15], has been adopted in this study, as it is more information-intensive and helps improve the efficiency of the WTP estimation. Moreover, this study also examines the significant determinants of visitors’ WTP, including the socio-economic characteristics and the attitudinal variables. Therefore, results from this study help create informed pricing policies for China’s national parks to prevent negative public opinions about the entrance fees and to meet social demands.

## 2. Theoretical Framework

The economic mechanism of setting a fee-paying policy is that visitors benefit from the use and non-use values generated from the national parks [12], which provides park authorities with the basis for establishing entrance fees to supplement government budget funds [7,8]. According to previous studies [5,6,9,10,11,12], WTP is the most frequently used indicator for the economic valuation of the various benefits related to the national parks. Economic theory indicates that the higher the entrance fee of a national park, the fewer visitors that are willing to pay it. Suppose the number of visitors decreases significantly after a relatively high entrance fee is introduced. In that case, this will defeat the purpose of establishing national parks to promote the public’s access to outdoor recreational opportunities and may also reduce the potential revenue due to the reduction in the number of visitors. Therefore, it is meaningful to estimate and aggregate a particular national park’s WTP to form a demand curve for consumer surplus analysis at different entrance fee levels [5].

To study visitors’ WTP entrance fees for the national parks, the contingent valuation method (CVM), consistent with neoclassical economics, has been one of the most widely used methods [15]. The CVM involves asking people’s purchase decisions on an amenity at the stated prices in a hypothetical marketplace and has been widely applied to the analyses of various protected areas, and Halkos et al. [14] and Peters & Hawkins [16] gave a review of these studies including the urban green spaces, forest parks, marine parks, and wildlife parks. The WTP estimated from CVM information can play an important role in the decision-making process related to conserving environmental resources to maximize society’s welfare. The essential part of the CVM study is to elicit respondents’ WTP values in a hypothetical scenario that depicts the changes and improvements of the studied environmental goods and services. Compared with other WTP elicitation techniques of the CVM (such as open-ended questions, a payment card, and an iterative bidding approach), the dichotomous-choice format is the most widely used technique since it has the advantage of mimicking the behaviour in regular markets and thus encouraging survey respondents to make incentive-compatible responses [17].

The literature has analyzed the differences in people’s WTP for recreational parks from the aspect of certain demographic and attitudinal factors. As for the demographic factors, a number of studies have found that income and education positively affect WTP, while age is negatively related to WTP [18,19]. Previous studies found the influences of gender [5,9,20,21], place of residence [5,22], visiting experience [23,24], and staying time [4,25] could be positive, negative, or insignificant, and need to be tested further. As for the attitudinal factors, some researchers have studied the influence of fairness, equity, and trust on WTP [4,5,6,26,27], whereas only a few researchers have attempted to empirically test the combined effects on recreational parks’ WTP using both demographic and attitudinal factors [4,5,6].

The previous studies have contributed to building a conceptual framework for estimating people’s WTP for national parks and understanding the differences in people’s WTP. This study is based on previous empirical evidence. We adopt the DBDC format of the CVM [28] to elicit visitors’ WTP for China’s newly established national parks and examine the significant determinants, both socio-economic and attitudinal, as factors of WTP.

## 3. Materials and Methods

### 3.1. Study Area

The area of focus for this study is the planned Qinling National Park (QNP), located in the core area of the Qinling Mountains in Shaanxi Province, China (Figure 1). As a natural boundary between China’s north and south and a dividing line between China’s warm temperate zone and subtropical zone, the Qinling Mountains houses a wide variety of wild biology, including more than 700 vertebrates and over 3800 seed plant species. At present, the Qinling Mountains have numerous geographically dispersed protected areas, including nature reserves, parks, and tourist resorts. The fragmented protected areas make it difficult to coordinate and unify the protection policies for various species and create barriers to scientific research. Against the background of China’s plan to establish national parks [2], the preparations to build the QNP started from the layout of the *General Plan for the Establishment of the QNP (2017-2026)* in 2016. Since then, several preparations for establishing the QNP, including appointing an expert committee, formulating the general plan, and submitting the application, have been actively carried out. On 25 October 2021, the establishment of the QNP, planned to be completed by 2025, received formal approval from China’s National Park Administration.

The proposed QNP (Figure 1) covers over 17 thousand square kilometres with rich wildlife and biodiversity resources. This area is home to 95% of endangered animals and plants in Shaanxi province. In particular, the QNP contains 13 species of wild animals and five species of plants under National Class I protection. Among the protected animals, giant panda, takin, golden monkey and crested ibis are typical "flagship species," known as the "four treasures of the Qinling Mountains." According to the establishment plan, a unified management system, the Qinling National Park Administration, will be founded to integrate the management functions of 79 protected areas—24 nature reserves, 35 forest parks, 7 wetland parks, 8 historical and scenic sites, 4 geological parks, and a portion of the pilot Giant Panda National Park located in Shaanxi province. The establishment of QNP will replace the segmented and overlapped management departments with a unified and efficient management system for natural resource protection. The protected area under the QNP will be divided into two functional zones with different constraints: the core protection zone, where human activities are prohibited, covering 55% of the park; and the general control zone, where human activities are restricted, covering 45% of the park.

As a popular tourist destination in China, the Qinling Mountains are famous for their extraordinary beauty and abundant cultural and historic attractions. The newly established QNP will continue to provide the public with access to areas for tourism, education, and entertainment. Thus, a survey to elicit tourists’ WTP for the future established QNP could provide information to improve public acceptance because it may allow the involvement of public opinions in the early stage of the management process.

### 3.2. Survey Instrument

The questionnaire was designed after the pretest and the pilot test to ensure its validity and user-friendliness. The pretest was conducted with eight experts in the field of ecology and economics to develop and improve the questionnaire. After that, a pilot test with a total of 50 respondents was carried out in a series of interviews with focus groups who helped to ensure the precision and clarity of all of the questions. The final version of the questionnaire started with introducing the respondents to the academic purpose of the survey to reduce rejection rates and obtain accurate information. Our questions were structured with four sections that helped investigate tourists’ WTP for the QNP’s entrance fees and determine the predictors of the WTP.

The first section consisted of prelude general questions about the respondents’ experience and attitudes of the tourist and resort zones of the Qinling Mountains. Respondents’ travel experience proved to be influential variables in explaining recreation activity in previous studies [16,17,18,19,20,21,22,23,24,25,26,27,28,29,30]. Thus, the travel experience variables in our questionnaire involved visitors’ duration of stay, visitor group, previous visit experience to the Qinling Mountains, and intention of a future visit to the Qinling Mountains. Attitudinal-related questions were adapted from the literature [5] using the five-point Likert scale format. They measured the respondents’ attitude towards the commercial exploitation of natural resources and the current management conditions. After these general relaxing questions, WTP and socio-economic questions were introduced in later sections.

The second core section presented value eliciting questions concerning the respondents’ WTP for the QNP’s entrance fee. The elicitation questions had been designed as a double bounded dichotomous choice (DBDC) format of the CVM method. The payment vehicle for WTP questions was an entrance fee to the planned QNP due to the fact that visitors were already used to paying park entrance fees. This part began with a brief introduction to the planned QNP with a map (Figure 1), and details to explain the improved protected management area. It was followed by the DBDC elicitation format, where respondents were offered two sequential dichotomous choice questions to reveal their WTP. For doing so, each respondent was firstly offered an initial bid value for the QNP’s entrance fee and was asked to provide a dichotomous choice response (i.e., “Yes” or “No”). If the respondent volunteered to pay the first bid amount, the second dichotomous question would be offered with a higher bid value and vice versa.

The third section was a follow-up section that helped improve the WTP estimations for the QNP’s entrance fee by reducing the unidentified biases. Respondents were asked to state the main reason for being unwilling to pay if they refused to pay for the entrance fee in two consecutive dichotomous choice questions. Those reasons included “It is not my responsibility to pay,” “Do not believe may payment will be used for nature conservation,” “The fee is too expensive,” and “I am not interested in the QNP.” To reduce the bias for valuation, the respondents who gave the first two reasons would be excluded from the analysis on the basis of being protesters [31]. Moreover, respondents were asked to state the maximum amount they could afford for the QNP’s entrance fee. To ensure consistency from our WTP questions, we removed the respondents from our sample if the stated fees from open-ended questions were beyond the WTP intervals determined by previous DBDC questions.

The final section recorded the socio-economic characteristics of the respondents, which had been considered as significant predictors of the WTP in most CVM studies (as we explained in Section 2). These variables included gender, age, province of residence, level of education, monthly income per person in the family, and the number of trips on average per year to natural attractions.

### 3.3. Data Collection

The target population of our analysis has been potential visitors possessing the ability to pay. Since the QNP was still in the planning and construction stage, information as to the demographic composition of the target population was not available. Thus, probabilistic sampling methods could not be used in this study. Instead, we collected our onsite data from visitors accessing the main tourist and resort zones of the planned QNP, including three National AAAAA Level Tourist Attractions (Taibai Mountain National Forest Park, Huashan Scenic Area, and Jinsi Gorge Scenic Area) and three National AAAA Level Tourist Attractions (Niubeiliang National Forest Park, Foping National Nature Reserve, and Cuihuashan National Geopark). These areas were selected by considering the difference in city locations, popularity, sizes of the scenic area, and typical recreational activities (e.g., scenic beauty, recreation, and wildlife watching).

The full survey was implemented during the tourism season of June to September 2021. Face-to-face interviews were conducted to ensure the quality and integrity of our collected data. The interviews were carried out by trained investigators who would explain the objectives and details of the survey to the respondents, which helped reduce any potential miscommunication and strategic bias. Respondents were selected through systematic random sampling. One in every five visitors (aged 18 and above) who showed up at the study area was approached to participate in the survey, which was estimated to last about 10 minutes. For group visitors of having companies, the individual with the minimum last four digits of telephone number was selected. At the end of the survey, respondents would receive souvenirs as a thank-you gift.

A total of 1335 questionnaires were collected from the 1600 approached visitors (response rate nearly 90%), of which 1307 were usable (87.1% of the total sample) with complete information. Further analysis based on the third part of our questionnaire indicated that 76 responses were protesters (23 selected "not their responsibility" and 53 selected "do not believe"), and 39 responses had inconsistency problems. After removing the invalid responses, the remaining sample comprised 1215 valid questionnaires with an 81.0% effective rate.

### 3.4. Model Specification

As indicated in the survey instrument, the model specification will include three types of variables: the socio-economic characteristics denoted as **Socio**, the attitudinal variables denoted as **Att**, the bid values denoted as **bid**.

The bid values (*bid*) ranged from 50 yuan to 300 yuan (1 U.S. dollar equals around 6.4 yuan during the survey) based on the current ticket prices of the tourist attractions and our pilot study. In the DBDC format of CV responses, the initial bid, denoted as *bid_i_*, was randomly selected from a bid set, (*bid_L_*, *bid_i_*, *bid_H_*), where *bid_L_* < *bid_i_* < *bid_H_*. Accordingly, we divided the range of entrance fee into four bid sets, i.e., (50, 100, 150), (100, 150, 200) (150, 200, 250), and (200, 250, 300). During the survey, respondents were asked twice whether they would be willing to pay the entrance fee of the future established QNP at a prespecified bid level. The first bid level would be the middle value of each bid set, *bid_i_*. If a negative answer is given in the first bid, a lower bid level (*bid_L_*) within the same bid set, otherwise a higher bid (*bid_H_*), would be proposed in the follow-up question.

According to Cameron [32] and Hanemann [33], the utility of respondent *n*, denoted as *U_n_*, consists of the observable linear part *V*, and the unobservable stochastic part *ε*:(1)$$U_{n}=V_{n}\left(Q,y,Att,Socio\right)\;+\;\varepsilon =\alpha *Q+\beta *y+\Gamma *Att+\Theta *Socio+\varepsilon$$
where *Q* is the quality of protected areas, with Q=0 indicating the status quo (i.e., current protected area system) and Q=1 indicating the changed scenario (i.e., the establishment of QNP); and *y* is the personal income. If the respondent answers “yes” to a pre-determined bid value *bid*, given the change in the quality of natural sites, we can obtain the following utility difference function:(2)$$V_{n}\left(1,y-bid,Att,Socio\right)\;-\;V_{n}\left(0,y,Att,Socio\right)\;>\varepsilon_{0}-\varepsilon_{1}$$

Given the normal distribution of the stochastic part ε, $\varepsilon \sim N\left(0,\;\sigma^{2}\right)$ (Lopez-Feldman, 2010), the probability of a respondent answering “yes” would be:(3)$$\begin{array}{cc} {\mathrm{Prob}}_{n}\left(yes\right) & =1-1/\{1+\exp[\beta *\left(\frac{\alpha }{\beta }-bid+\frac{\Gamma }{\beta }*Att+\frac{\Theta }{\beta }*Socio\right)]\}\\ & =1-G\left(A\right) \end{array}$$
where α/β is the WTP for general quality change of natural sites; Γ/β and Θ/β is the WTP variations associated with Att and Socio, respectively.

In the DBDC format of CV responses, there are four possible combinations of the response sequences with different probabilities: (i) the probability of answering yes-yes for *bid_i_* and *bid_H_*, denoted as $P_{n}^{YY}$; (ii) the probability of answering yes-no for *bid_i_* and *bid_H_*, denoted as $P_{n}^{YN}$; (iii) the probability of answering no-yes for *bid_i_* and *bid_L_*, denoted as $P_{n}^{NY}$; (iv) the probability of answering no-no for *bid_i_* and *bid_L_*, denoted as $P_{n}^{NN}$. Consequently, the log-likelihood function for the DBDC model, referring to Hanemann et al. [34], is as follows ($I_{n}$ is the binary-valued indicator variables corresponding to each of the possible combinations):(4)$$\ln L=\overset{N}{\underset{n=1}{\sum }}\left\{\begin{array}{c} I_{n}^{YY}*\ln P_{n}^{YY}\\ +I_{n}^{YN}*P_{n}^{YN}\\ +I_{n}^{NY}*P_{n}^{NY}\\ +I_{n}^{NN}*P_{n}^{NN} \end{array}\right\}=\overset{N}{\underset{n=1}{\sum }}\left\{\begin{array}{c} I_{n}^{YY}\ln\left[1-G\left(bid^{u}\right)\right]\\ +I_{n}^{YN}\ln\left[G\left(bid^{u}\right)\;-\;G\left(bid^{l}\right)\right]\\ +I_{n}^{NY}\ln\left[G\left(bid^{l}\right)\;-\;G\left(bid^{u}\right)\right]\\ +I_{n}^{NN}\ln\left[G\left(bid^{l}\right)\right] \end{array}\right\}$$

## 4. Results

### 4.1. Descriptive Statistics

The descriptive statistics of the 1215 valid questionnaires are presented in Table 1. The socio-economic variables used in the model specification have been the respondent’s *gender*, *age*, *residence*, *education*, *income*, *trip*, *stay*, *alone*, and *visit*. The respondents’ gender composition was about 60%:40% male and female, respectively. Their average age was approximately 35, and the average education level was nearly some college education. The visitors’ characteristics of being young and highly educated were consistent with previous studies conducted in other protected areas [35,36]. Most (81.6%) visitors resided in Shaanxi Province because the confirmed cases of COVID-19 discouraged visitors’ from travelling to other provinces. The incomes were classified into six brackets, and on average, participants indicated that their average household income ranged from 2000 yuan to 4000 yuan. The annual average trips of our respondents ranged from four to six times. In addition, the average days of stay in the Qinling Mountains was 1.6. Most visitors travelled with a company (63.0%), and had the visiting experience to Qinling Mountains (74.6%). Our sample of the visitors was representative since the socio-economic characteristics are consistent with analogous surveys conducted in Shaanxi province [37], as well as in other tourist attractions of China’s pilot national parks [35,36].

As indicated in Section 3.2, attitudinal variables included intention of a future visit to Qinling Mountains (*intention*), degree of concern about the commercial exploitation of natural resources in Qinling Mountains (*concern*), and degree of satisfaction about the current management conditions (*satisfaction*). Table 1 shows that respondents had the intention to visit Qinling Mountains in the future (84.3%). In addition, our respondents gave a mean score on *concern* and *satisfaction* greater than 3. Therefore, we can infer that those visitors were generally satisfied with the current management conditions, but they would prefer to see less commercial development in the natural areas. 

As for the biding characteristics, Table 1 also indicates that 59.3% of the respondents were willing to pay an entrance fee for the first amount proposed, and this percentage decreased slightly to 55.8% for the second amount proposed; 38.7% answered “yes” to all the dichotomous choice questions, and 23.5% answered “no” to all dichotomous choice questions.

### 4.2. Model Estimation Results

The specified model for a DBDC format has been estimated with the Stata “doubleb” command developed by Lopez-Feldman [28], and the results are shown in Table 2. Table 2 presents the estimated WTP for each of the explanatory variables (i.e., α/β, **Γ**/β, and **Θ**/β in Equation (3)) and contains three models. Model 1 is the constant-only model, which predicts the WTP from the basic regression. Model 2 estimates the marginal WTP of all the explanatory variables. Model 3 identifies the statistically significant predictors with the backward elimination process, i.e., iteratively removing the least significant variable until all the covariates are significant.

Model 1 shows that the constant variable is statistically significant at the 1% level. It indicates that the establishment of the QNP is essential for our respondents. The average entrance fee that our respondents are willing to pay is about 200 yuan (from basic regression analysis). After considering the covariates, Model 2 and Model 3 improved the model fit significantly since the L.R. chi2 statistics are both significant. The likelihood-ratio test has also been performed to compare the fit of Model 2 to the fit of Model 3. The resulting statistic (2.82 with degrees of freedom of 5) is not significant (p-value equals 0.73), which indicates that removing the non-significant covariates does not lead to the change in model fit. Thus, Model 2 and Model 3 have no difference but are significantly better than Model 1 in model fit.

The comparison between Model 2 and Model 3 shows that the significance and magnitude of the common covariates are resistant to model specifications, which indicates the robustness in data analysis. Moreover, less than half of the predictors have no significant influences on WTP and have been removed for Model 3. Previous studies found that age may show a positive [4] or a negative [5] effect on WTP. However, our study does not find a well-defined relationship with respondents’ WTP, although the negative sign may indicate that younger respondents were more receptive to paying for the QNP. The influence of *residence* is insignificant, indicating no difference in WTP between visitors living in and outside Shaanxi province. This study also finds no significant difference in WTP between visitors with or without a company, with or without visiting experience, and with different satisfaction levels.

Among the significant socio-economic variables (**Socio**), *gender*, *edu*, *income*, *trip,* and *stay* positively affect the WTP. While some studies like Reynisdottir et al. [5] and Opačak & Wang [20] found an insignificant or negative influence of females on WTP, our study found that females have a higher WTP ( 24.5 yuan) than males, which is in line with the studies of López-Mosquera [21] and Piriyapada & Wang [9]. A possible explanation is that females tend to have stronger pro-environmental attitudes [38,39], which leads to a stronger WTP for nature conservation. As expected, the coefficients of education and income are significantly positive. The positive signs suggest that wealthier and better-educated respondents would be willing to pay more entrance fees for the QNP. Similar results could also be found in most WTP studies of natural attractions such as Samdin et al. [18] and Bhat et al. [19]. Regarding the influence of *trip*, this has a positive effect on WTP. The possible reason is that visitors who have more trip experience might be more accustomed to paying entrance fees to natural attractions and thus would be more willing to pay for the QNP. Contrary to the study of Wang and Jia [4], but in agreement with Bhandari & Heshmati [25], we find that respondents’ WTP increased with longer staying time. The possible reason is that visitors became less sensitive to the entrance fee since the share of entrance fee to total travel cost declined with longer stays.

Among the significant attitudinal variables (**Att**), *intention* and *concern* positively impact WTP. It is interesting to find a significant and positive coefficient on the intention of future visit variable, considering that the previous visit experience has no significant influence on WTP. The possible reason is that visitors having future visiting intentions would be willing to pay more if they were attracted by the unique appeal of Qinling Mountains for its abundant natural, cultural and historic attractions. Visitors who are more concerned with commercial exploration will have a higher WTP. The possible reason is that they would prefer to change the current situation, and the establishment of QNP helps protect the natural ecosystems authentically and integrally.

### 4.3. WTP Estimation Results

The WTP estimation of the entrance fee can be used as a reference to set the pricing policy for the QNP. Table 3 presents the mean WTP and its summary statistics from each of the estimated double bounded models. The mean WTP has been calculated by the weighted sum, multiplying each variable’s mean by its marginal WTP and adding the results of the significant predictors.

Table 3 shows that our estimated mean WTP has consistency across all three models. Since the 95% WTP intervals from different models overlap each other, there is no significant difference with the estimated mean WTPs in different models. The L.R. tests among competing models indicate that Model 3 is the preferred model, and results obtained from Model 3 will be considered the final results in our study. Thus, the mean WTP value is nearly 200 yuan with a 95% confidence interval of 193.1 yuan to 203.3 yuan.

The visitor’s demand curve can be reflected by aggregating each respondent’s WTP, i.e., the weighted sum of predictors at the individual level. Figure 2 shows the demand curve, which is close to being a reflection of the S-shape in the y-coordinate. As expected, the demand curve slopes downwards, indicating that the visiting rate will decrease with increased entrance fees. The total number of visitors to the QNP is expected to be 96 million after the establishment in 2025 [40]. Based on our estimation, the expected ticket revenue (i.e., the product of ticket price, expected number of visitors, and visiting rate) curve has been realized and is presented in Figure 2. The demand curve becomes inelastic when the entrance fee is about 160 yuan. Meanwhile, the expected ticket revenue would reach its maximum of 13.8 billion yuan given the entrance fee of 160 yuan.

At the recommended entrance fee of 160 yuan, our respondents can be divided into two groups based on the predicted individual’s WTP, i.e., respondents whose WTP is above 160 yuan (“group accept” in Figure 3) and bellow 160 yuan (”group reject” in Figure 3). The differences in group means in Figure 3 is consistent with the results of Model 3 in Table 2, and all group means of the “group accept” is higher than that of the ”group reject”.

The recommended price of the QNP’s entrance fee, 160 yuan, is within the ticket price range (90 yuan to 180yuan) of Shaanxi province’s AAAAA level tourist attractions, which increases the credibility of our research. Considering that China’s per capita GDP and per capita disposable income in 2020 were respectively 72,447 yuan and 32,189 yuan, our estimated entrance fee accounts for 0.22% and 0.50% of the per capita GDP and per capita disposable income, respectively. The GDP proportion (0.22%) is far larger than that for the average entrance fee of national parks in the U.S. (0.022%), Australia (0.015%), and Canada (0.010%) [41]. This accounts for the fact that the funding of China’s tourist attractions relied heavily on entrance fees [4], and visitors have become accustomed to the high-priced nature reserves due to cultural and historical norms [16]. 

## 5. Conclusions

This study analyses visitors’ WTP for the national park’s entrance fee against the background of China’s rapid progress in developing the national park system. In the process, the double-bounded dichotomous choice format of the CVM has been utilised to obtain the WTP estimates by taking the planned QNP as our study area. This study collected field survey data that will be helpful for authorities to determine the most appropriate entrance fee level based on visitors’ demand curve. A total of 1215 valid questionnaires were used for our empirical analysis. The results show that the mean WTP is around 200 yuan, and the recommended entrance fee to the QNP is 160 yuan, as this is the point at which the expected ticket revenue will reach its maximum of 13.8 billion yuan. Thus, our study confirms that visitors are ready and willing to pay an entrance fee for the planned QNP, and the establishment of the QNP will generate a significant number of economic benefits. Accordingly, several policy recommendations can be suggested. The expected revenue from entrance fees will be a vital capital source of the QNP’s biodiversity conservation and environmental protection. It is essential to compare the expected revenue with the cost of operating a fee collection system to provide the economic justification of an obligatory entrance fee programme. This research also shows the visiting rate change and the corresponding change in entrance fees, which helps balance visiting opportunities and nature conservation goals.

Moreover, our study also reveals the influential factors of visitors’ WTP an entrance fee for the QNP. Our study demonstrates that females have more WTP for the QNP’s entrance fee than males. Besides, education level, income, and the number of trips to natural attractions positively affect WTP. It should be noted that studies on the socio-economic variables’ influences on WTP vary greatly in their context and our findings’ comparability is therefore limited to our specific study context. This study is also significantly influenced by the experiential and attitudinal predictors. Visitors’ duration of stay, the intention of a future visit, and concern about the commercial exploitation positively impact WTP. Accordingly, park managers and policymakers need to understand the individual differences in WTP to promote social marketing campaigns linked to the national park. Authorities of the QNP should also plan to improve the visiting experience, including the unique opportunities for recreation, research, and education, in order to increase the likelihood of visitors’ WTP the entrance fee.

Finally, the CVM serves as a valuable tool in providing relevant information for setting the QNP’s entrance fee levels. The results of this study on visitors’ WTP and methods for determining an appropriate entrance fee level are also applicable to other pilot national parks in China. Meanwhile, there are several limitations of this study. Firstly, although the COVID-19 pandemic was well controlled during the data collection period, the proportion of visitors from foreign countries and outside Shaanxi province has slumped significantly. Thus, a future research direction would be the comparison of WTP among different visitor groups from local Shaanxi province, other provinces of China, and foreign countries. Results from this comparison would be helpful to make price discrimination strategies. Secondly, the WTP during the pandemic and in the post-pandemic future could also be compared to find whether the pandemic might change visitors’ preferences for national parks. Thirdly, there is also a need for understanding visitors’ decision-making processes of WTP [38,42] (e.g., attitude and social norm), which were beyond the scope and purposes of this study, in order to help policymakers develop more effective marketing strategies to manage the park better.

## Figures and Tables

**Figure 1 ijerph-18-13410-f001:**
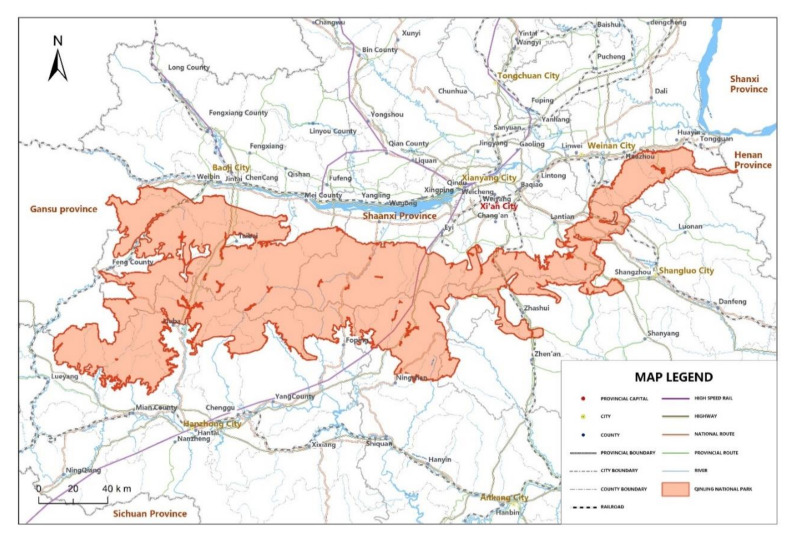
The map of the proposed Qinling National Park.

**Figure 2 ijerph-18-13410-f002:**
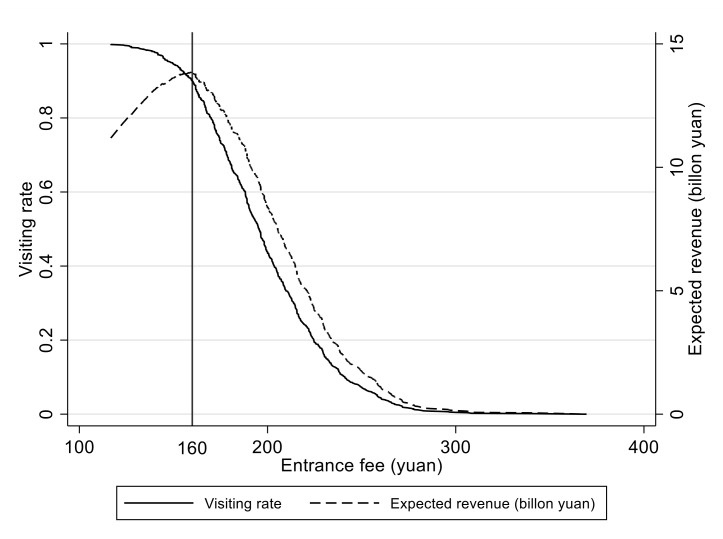
Demand curve and expected revenue.

**Figure 3 ijerph-18-13410-f003:**
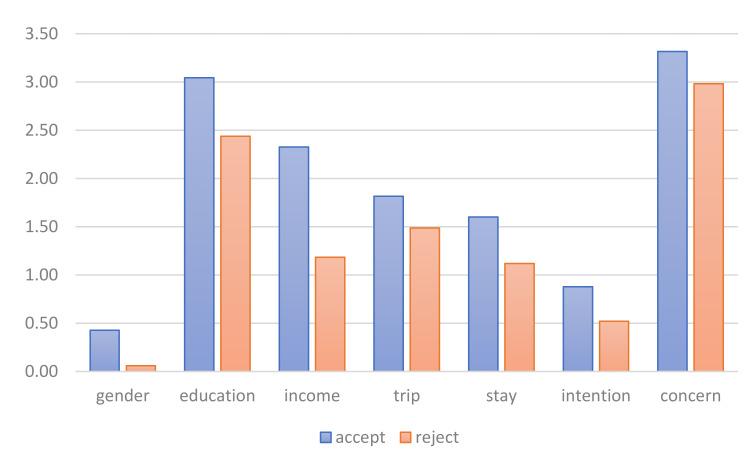
Comparing the group means of the respondents’ Characteristics.

**Table 1 ijerph-18-13410-t001:** Socio-economic, attitudinal, and biding characteristics of the respondents.

Variables	Description	Statistics	
gender	Male = 0	Frequency	739
	Female = 1	Frequency	476
age	Age of the respondents divided into five categories	Mean	2.406
		Std. dev.	1.073
	From 18 to 25 = **1**	Frequency	267
	From 26 to 35 = **2**	Frequency	449
	From 36 to 45 = **3**	Frequency	262
	From 46 to 55 = **4**	Frequency	213
	Above55 = **5**	Frequency	24
education	Education level of the respondents divided into four categories	Mean	2.984
		Std. dev.	0.83
	Junior school or under = **1**	Frequency	64
	Senior school = **2**	Frequency	236
	Some college = **3**	Frequency	570
	Post graduate school = **4**	Frequency	345
*residence*	Shaanxi province = **0**	Frequency	992
	Outside shaanxi province = **1**	Frequency	223
income	Monthly per-capita income (in yuan) in the family divided into five categories	Mean	2.215
		Std. dev.	1.219
	Bellow 2000 = **1**	Frequency	467
	From 2000 to 4000 = **2**	Frequency	277
	From 4000 to 6000 = **3**	Frequency	289
	From 6000 to 8000 = **4**	Frequency	107
	Above 8000, +∞ = **5**	Frequency	75
*trip*	Annual average number of trips to natural attractions divided into three categories	Mean	1.784
		Std. dev.	0.686
	Less than 3 trips = **1**	Frequency	446
	4 to 6 trips = **2**	Frequency	586
	More than 6 trips = **3**	Frequency	183
*stay*	The expected number of days of stay	Mean	1.554
		Std. dev.	0.942
*alone*	Travel with others = **0**	Frequency	450
	Travel alone = **1**	Frequency	765
*visit*	Have no visit experience to Qinling Mountains = **0**	Frequency	309
	Have visit experience to Qinling Mountains = **1**	Frequency	906
*intention*	Have no intention of a future visit = **0**	Frequency	191
	Have intention of a future visit = **1**	Frequency	1024
*concern*	Concern about the commercial exploitation of natural resources in five-point Likert scales	Mean	3.283
		Std. dev.	1.289
	Not at all = **1**	Frequency	141
	Slightly = **2**	Frequency	171
	Somewhat = **3**	Frequency	397
	Moderately = **4**	Frequency	215
	Extremely = **5**	Frequency	291
*satisfaction*	Level of the government trust in five-point Likert scales	Mean	3.765
		Std. dev.	1.223
	Very untrusted = **1**	Fren	101
	Relatively untrusted = **2**	Frequency	86
	Average = **3**	Frequency	220
	Relatively trusted = **4**	Frequency	398
	Very trusted = **5**	Frequency	410
*bid1*	The initial bid randomly assigned to the respondent in yuan	Mean	175.638
		Std. dev.	56.27
*bid2*	The second bid assigned following the initial bid in yuan	Mean	184.979
		Std. dev.	55.344
*answer*	“yes” and “yes” for the sequential questions	Frequency	470
	“yes” and “no” for the sequential questions	Frequency	251
	“no” and “yes” for the sequential questions	Frequency	208
	“no” and “no” for the sequential questions	Frequency	286

**Table 2 ijerph-18-13410-t002:** Estimation results of the double bounded models.

Variables	Model 1	Model 2	Model 3
*Constant*	197.951 *** (2.762)	43.513 ** (19.878)	38.772 ** (16.646)
*gender*		24.541 *** (5.300)	24.547 *** (5.306)
*age*		−1.481(2.398)	
*education*		11.282 *** (3.137)	10.923 *** (3.130)
*residence*		7.719(8.760)	
*income*		17.406 *** (2.193)	17.310 *** (2.191)
*trip*		9.501 ** (3.995)	10.000 ** (3.942)
*stay*		14.596 *** (3.811)	16.554 *** (3.104)
*alone*		0.271(5.324)	
*visit*		−6.774(5.912)	
*intention*		25.993 *** (7.099)	25.939 *** (7.104)
*concern*		4.049 ** (2.005)	4.091 ** (2.002)
*satisfaction*		1.317(2.117)	
log likelihood	−1442.945	−1365.069	−1366.482
L.R. chi2	-	155.75 ***	152.93 ***

Notes: Definition of the variables are shown in Table 1. The standard error of each estimated coefficient is presented in the parenthesis. *, ** and *** denote the significance at 10%, 5% and 1% levels, respectively. L.R. chi2 is the result of a likelihood-ratio test statistic, distributed chi-squared, to test the difference in model fit.

**Table 3 ijerph-18-13410-t003:** Mean willingness to pay according to the double-bounded models.

Mean WTP	Mean	Std. Err.	*p*-Value	95% Conf. Interval
Model 1	197.951	2.762	0.000	from 192.538 to 203.365
Model 2	200.176	11.448	0.000	from 177.739 to 222.613
Model 3	198.176	2.607	0.000	from 193.067 to 203.286

Notes: Std. Err. is the standard error of the estimated mean WTP; *p*-value indicates the significant level; 95% Conf. Interval is the WTP interval associated with the 95% confidence level.

## Data Availability

The datasets used and analyzed during the current study are available from the corresponding author on reasonable request.

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
