# Peer review of "Visitor’s Willingness to Pay for National Park Entrance Fees in China: Evidence from a Contingent Valuation Method"

_ijerph, 2021, doi:10.3390/ijerph182413410_

Round 1

Reviewer 1 Report

The proposed research may have a high interest for the management and conservation of natural areas by public managers in China.

However, the article presents some deficiencies that must be corrected prior to the publication of the article.

The most evident of all is the lack of a theoretical framework that, based on a review of the scientific literature, can support the construction of hypotheses and, at the same time, support and argue the results and conclusions.

On the other hand, in addition to providing the data in Table 1, it would be convenient to include a sociodemographic profile in absolute terms.

Finally, it should be noted that the conclusions refer to the existence of differences in terms of the WTP of men and women, or based on the level of education or income, conclusions that are not supported either methodologically or statistically.

Reviewer 2 Report

The article aims to study the willingness of visitors to national parks to pay an entrance fee. The idea of the article is interesting, although it is quite particular to the geographical area chosen for the study, in this case China. 

The article is written with a suitable research quality, dealing adequately with the main requirements that a research paper should have. 

However, I would like to comment on some circumstances that call attention and that could be improved. In the first place, it is noteworthy that the article lacks a theoretical framework, which is not obligatory, but it is linked to the fact that the first introductory point is scarce. If the theoretical framework is not to be added, it would be appropriate to expand the introduction and better contextualize the study to be carried out and the objective pursued.

The methodology in general is suitable and shows a good command of it. I would like to make some clarifications, especially in the conduct of the survey. In the text it is not entirely clear what type of variables have been chosen, why these variables have been chosen and what previous studies have dealt with these variables that are subsequently transformed into questions for the questionnaire. More specifically, I believe that point 2.2 Survey Instrument should be expanded with this information, and a table could even be added to include the variables to be studied and the authors/references that have dealt with these variables in their studies.

The results are adequate.

The conclusions are suitable but I have missed some comparisons with the results of other similar articles that would help to clarify a little more the benefits of the study.

Reviewer 3 Report

Authors described that visitors’ demand becomes 16
relatively inelastic at the recommended entrance fee of 160 yuan, showing the graph. It is ok, but if you revealed difference among gender, education, income levels at this price line, it would be more understandable. 

Round 2

Reviewer 1 Report

The introduction of the section corresponding to the theoretical framework is appreciated, although it could be somewhat more extensive and deep.